# Indiscriminate Data Poisoning Attacks on Neural Networks[*]

**Yiwei Lu, Gautam Kamath[†], Yaoliang Yu**
{yiwei.lu, gckamath, yaoliang.yu}@uwaterloo.ca
University of Waterloo

## Abstract

Data poisoning attacks, in which a malicious adversary aims to influence a model by injecting "poisoned" data into the training process, have attracted significant recent attention. In this work, we take a closer look at existing poisoning attacks and connect them with old and new algorithms for solving sequential Stackelberg games. By choosing an appropriate loss function for the attacker and optimizing with algorithms that exploit second-order information, we design poisoning attacks that are effective on neural networks. We present efficient implementations by parameterizing the attacker and allowing simultaneous and coordinated generation of tens of thousands of poisoned points, in contrast to existing methods that generate poisoned points one by one. We further perform extensive experiments that empirically explore the effect of data poisoning attacks on deep neural networks. Our paper set up a new benchmark on the possibility of performing indiscriminate data poisoning attacks on modern neural networks.

## 1 Introduction

Adversarial attacks have repeatedly demonstrated critical vulnerabilities in modern machine learning (ML) models [25, 34, 21]. As ML systems are deployed in increasingly important settings, significant effort has been levied in understanding attacks and defenses towards *robust* machine learning.

In this paper, we focus on *data poisoning attacks*. ML models require a large amount of data to achieve good performance, and thus practitioners frequently gather data by scraping content from the web [11, 37]. This gives rise to an attack vector, in which an adversary may manipulate part of the training data by injecting poisoned samples. For example, an attacker can *actively* manipulate datasets by sending corrupted samples directly to a dataset aggregator such as a chatbot, a spam filter, or user profile databases; the attacker can also *passively* manipulate datasets by placing poisoned data on the web and waiting for collection. Moreover, in *federated learning*, adversaries can also inject malicious data into a diffuse network [30, 22].

A spectrum of such data poisoning attacks exists in the literature, including *targeted*, *indiscriminate* and *backdoor* attacks (see Appendix A for a detailed comparison). We focus on indiscriminate attacks for image classification, where the attacker aims at decreasing the overall test accuracy of a model by adding a small portion of poisoned points. Current indiscriminate attacks are most effective against convex models [2, 18, 19, 31], and several defenses have also been proposed [32, 6]. However, existing poisoning attacks are less adequate against more complex non-convex models, especially deep neural networks, either due to their formulation being inherently tied to convexity or computational limitation. For example, most prior attacks generate poisoned points sequentially.

---

[*]GK and YY are listed in alphabetical order.

[†]Supported by an NSERC Discovery Grant, an unrestricted gift from Google, and a University of Waterloo startup grant.

2022 Trustworthy and Socially Responsible Machine Learning (TSRML 2022) co-located with NeurIPS 2022.

Thus, when applied to deep models or large datasets, these attacks quickly become computationally infeasible. To our knowledge, a systematic analysis on poisoning deep neural works is still largely missing—a gap we aim to fill in this work.

To address this difficult problem, we design more versatile data poisoning attacks by formulating the problem as a non-zero-sum Stackelberg game, in which the attacker crafts some poisoned points with the aim of decreasing the test accuracy, while the defender optimizes its model on the poisoned training set. We exploit second-order information and apply the Total Gradient Descent Ascent (TGDA) algorithm to address the attacker's objective, even on non-convex models.

Moreover, we address computational challenges by proposing an efficient architecture for poisoning attacks, where we parameterize the attacker as a separate network rather than optimizing the poisoned points directly. By applying TGDA to update the attacker model directly, we are able to generate tens of thousands of poisoned points simultaneously in one pass, potentially even in a coordinated way.

In this work, we make the following contributions: (1) We construct a new data poisoning attack based on TGDA that incorporates second-order optimization. In comparison to prior data poisoning attacks, ours is significantly more effective and runs at least an order of magnitude faster. (2) We propose an efficient attack architecture, which enables a more efficient, clean-label attack. (3) We conduct experiments to demonstrate the effectiveness of our attack on neural networks and its advantages over previous methods.

## 2 Total Gradient Descent Ascent Attack

In this section, we formulate the indiscriminate attack and introduce our attack algorithm. We first briefly introduce the Stackelberg game and then link it to indiscriminate data poisoning.

### 2.1 Preliminaries on Stackelberg Game

The Stackelberg competition is a strategic game in economics in which two parties move sequentially [36]. Specifically, we consider two players, a leader $\mathsf{L}$ and a follower $\mathsf{F}$ in a Stackelberg game, where the follower $\mathsf{F}$ chooses $\mathbf{w}$ to best respond to the action $\mathbf{x}$ of the leader $\mathsf{L}$, through minimizing its loss function $f$:

$$\forall \mathbf{x} \in \mathbb{X} \subseteq \mathbb{R}^d, \ \ \mathbf{w}_*(\mathbf{x}) \in \arg\min_{\mathbf{w} \in \mathbb{W}} f(\mathbf{x}, \mathbf{w}), \tag{1}$$

and the leader $\mathsf{L}$ chooses $\mathbf{x}$ to maximize its loss function $\ell$:

$$\mathbf{x}_* \in \arg\max_{\mathbf{x} \in \mathbb{X}} \ell(\mathbf{x}, \mathbf{w}_*(\mathbf{x})), \tag{2}$$

where $(\mathbf{x}_*, \mathbf{w}_*(\mathbf{x}_*))$ is known as a Stackelberg equilibrium.

For simplicity, we assume $\mathbb{W} = \mathbb{R}^p$ and the functions $f$ and $\ell$ are smooth, hence the follower problem is an instance of unconstrained smooth minimization.

### 2.2 On Data Poisoning Attacks

There are two possible ways to formulate data poisoning as a Stackelberg game, according to the acting order. Here we assume the attacker is the leader and acts first, and the defender is the follower. This assumption can be easily reversed such that the defender acts first. Both of these settings are realistic depending on the defender's awareness of data poisoning attacks. We will show in Section 4 that the ordering of the two parties affects the results significantly.

**Non-zero-sum formulation.** In this section we only consider the attacker as the leader as the other case is analogous. Here recall that the follower $\mathsf{F}$ (i.e., the defender) aims at minimizing its loss function $f$ under data poisoning:

$$\mathbf{w}_* = \mathbf{w}_*(\mathcal{D}_p) \in \arg\min_{\mathbf{w}} \ \mathcal{L}(\mathcal{D}_{tr} \cup \mathcal{D}_p, \mathbf{w}), \tag{3}$$

while the attacker aims at maximizing a different loss function $\ell$ on the validation set $\mathcal{D}_v$:

$$\mathcal{D}_{p_*} \in \arg\max_{\mathcal{D}_p} \mathcal{L}(\mathcal{D}_v, \mathbf{w}_*), \tag{4}$$

where the loss function $\mathcal{L}(\cdot)$ can be any task-dependent target criterion, e.g., the cross-entropy loss. Thus we have arrived at the bilevel optimization problem [23, 17, 19]:

$$\max_{\mathcal{D}_p} \mathcal{L}(\mathcal{D}_v, \mathbf{w}_*), \text{ s.t. } \mathbf{w}_* \in \arg\min_{\mathbf{w}} \mathcal{L}(\mathcal{D}_{tr} \cup \mathcal{D}_p, \mathbf{w}). \tag{5}$$

**Previous approaches.** While the inner minimization can be solved via gradient descent, the outer maximization problem is non-trivial as the dependence of $\mathcal{L}(\mathcal{D}_v, \mathbf{w}_*)$ on $\mathcal{D}_p$ is *indirectly* through the parameter $\mathbf{w}$ of the poisoned model. Thus, *applying simple algorithms (e.g., Gradient Descent Ascent) directly will result in zero gradient*. Nevertheless, we can rewrite the desired derivative using the chain rule: $\frac{\partial \mathcal{L}(\mathcal{D}_v, \mathbf{w}_*)}{\partial \mathcal{D}_p} = \frac{\partial \mathcal{L}(\mathcal{D}_v, \mathbf{w}_*)}{\partial \mathbf{w}_*} \frac{\partial \mathbf{w}_*}{\partial \mathcal{D}_p}$. The difficulty lies in computing $\frac{\partial \mathbf{w}_*}{\partial \mathcal{D}_p}$, i.e., measuring how much the model parameter $\mathbf{w}$ changes with respect to the poisoned points $\mathcal{D}_p$. Various approaches compute $\frac{\partial \mathbf{w}_*}{\partial \mathcal{D}_p}$ by solving this problem exactly via KKT conditions [2, 18], or approximately using gradient ascent [23].

**TGDA attack.** However, we can avoid such calculation using the Total gradient descent ascent (TGDA) algorithm [7, 8]: TGDA takes a total gradient ascent step for the attacker and a gradient descent step for the defender:

$$\mathbf{x}_{t+1} = \mathbf{x}_t + \eta_t \mathsf{D}_\mathbf{x} \ell(\mathbf{x}_t, \mathbf{w}_t), \tag{6}$$
$$\mathbf{w}_{t+1} = \mathbf{w}_t - \eta_t \nabla_\mathbf{w} f(\mathbf{x}_t, \mathbf{w}_t) \tag{7}$$

where $\mathsf{D}_\mathbf{x} := \nabla_\mathbf{x} \ell - \nabla_{\mathbf{wx}} f \cdot \nabla_{\mathbf{ww}}^{-1} f \cdot \nabla_\mathbf{w} \ell$ is the total derivative of $\ell$ with respect to $\mathbf{x}$, which implicitly measures the change of $\mathbf{w}$ with respect to $\mathcal{D}_p$.

We thus apply the total gradient descent ascent algorithm and call this the **TGDA attack**. Avoiding computing $\frac{\partial \mathbf{w}_*}{\partial \mathcal{D}_p}$ enables us to parameterize $\mathcal{D}_p$ and generate points indirectly by treating $\mathsf{L}$ as a separate model. Namely that $\mathcal{D}_p = \mathsf{L}_\theta(\mathcal{D}'_{tr})$, where $\theta$ is the model parameter and $\mathcal{D}'_{tr}$ is part of the training set to be poisoned. Thus, we have arrived a poisoning attack that generates $\mathcal{D}_p$ in a batch rather than individually, which greatly improves the attack efficiency.

**Zero-sum reduction.** Previous work [19] also proposed a reduced problem of Equation (5), where the leader and the follower consider the same loss function such that $f = \ell$:

$$\max_{\mathcal{D}_p} \ \min_{\mathbf{w}} \mathcal{L}(\mathcal{D}_{tr} \cup \mathcal{D}_p, \mathbf{w}), \tag{8}$$

This relaxation enables attack algorithms to optimize the outer problem directly. However, this formulation may be problematic as the training objective optimizes on poisoned points, which does not reflect its influence on test data.

This problem is addressed by [19] with an assumption that the attacker can acquire a *target* model parameter, usually using a label flip attack which considers a much larger poisoning fraction $\epsilon$. By adding a constraint involving the target parameter $\mathbf{w}_{tar}$, the attacker can search for poisoned points that maximize the loss $\ell$ while keeping low loss on $\mathbf{w}_*^{tar}$. However, such target parameters are hard to obtain since, as we will demonstrate, non-convex models appear to be robust to label flip attacks and there are no guarantees that $\mathbf{w}_*^{tar}$ is the solution of Equation (5).

## 3 An Efficient Attack architecture

In this section, we discuss how we use the TGDA attack to efficiently perform data poisoning attacks. We observe that existing data poisoning attacks have **two limitations**:

(1) **The first limitation**: existing attacks work under subtly different assumptions, on, for example, the attacker's knowledge, the attack formulation, and the training set size. These inconsistencies result in unfair comparisons between methods (see Table 5 in Appendix B for a detailed comparison).

Thus we set an experimental protocol for generalizing existing attacks and benchmarking data poisoning attacks for systematic analysis in the future. Here we fix three key variants:

*the attacker's knowledge:* as discussed in Appendix A, we consider training-only attacks (the attacker has access to training data $\mathcal{D}_{tr}$, the target model, and the training procedure);

*the attack formulation:* in Section 2, we introduce three possible formulations, namely non-zero-sum, zero-sum, and zero-sum with target parameters. We will show in the experiment section that the latter two would not work for neural networks.;

*the dataset size:* existing works measure attack efficacy with respect to the size of the poisoned dataset, where size is measured as a *fraction* $\epsilon$ of the training dataset. However, some works subsample and thus reduce the size of the training dataset. As we show in Figure 1, attack efficacy is *not* invariant to the size of the training set: larger training sets appear to be harder to poison. Furthermore, keeping $\epsilon$ fixed, a smaller training set reduces the number of poisoned data points and thus the time required for methods that generate points sequentially, potentially concealing a prohibitive runtime for poisoning the full training set. Thus we consider not only a fixed $\epsilon$, but also the complete training set for attacks.

(2) **The second limitation:** they approach the problem by optimizing individual points directly, thus having to generate poisoned points one by one.

Thus we design a new poisoning scheme that allows simultaneous and coordinated generation of $\mathcal{D}_p$ in batches requiring only one pass that involves three stages:

*Pretrain:* The goals of the attacker L are to: reduce the test accuracy (i.e., successfully attack) and generate $\mathcal{D}_p$ that is close to $\mathcal{D}_{tr}$ (i.e., thwart potential defenses). The attacker achieves the first objective during the attack by optimizing $\ell$. However, $\ell$ does not enforce that the distribution of the poisoned points will resemble those of the training set. To this end, we pretrain L to reconstruct $\mathcal{D}_{tr}$, producing a parameter vector $\theta_{pre}$. This process is identical to training an autoencoder.

For the defender, we assume that F is fully trained to convergence. Thus we perform standard training on $\mathcal{D}_{tr}$ to acquire F with $\mathbf{w}_{pre}$.

*Attack:* We generate poisoned points using the TGDA attack. We assume that the attacker can inject $\epsilon N$ poisoned points, where $N = |\mathcal{D}_{tr}|$ and $\epsilon$ is the power of the attacker, measured as a fraction of the training set size.

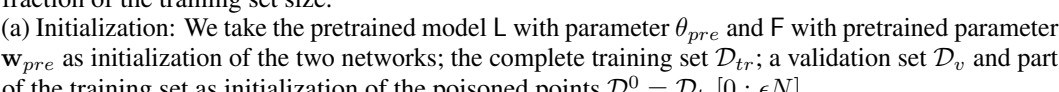

Figure 1: Comparing the efficacy of poisoning MNIST-17 with the PoisonSVM and Back-gradient attacks. The training set size is varied, while the ratio of the number of poisoned points to the training set size is fixed at $3\%$. These attacks become less effective as training set sizes increase.

(a) Initialization: We take the pretrained model L with parameter $\theta_{pre}$ and F with pretrained parameter $\mathbf{w}_{pre}$ as initialization of the two networks; the complete training set $\mathcal{D}_{tr}$; a validation set $\mathcal{D}_v$ and part of the training set as initialization of the poisoned points $\mathcal{D}_p^0 = \mathcal{D}_{tr}[0 : \epsilon N]$.

(b) TGDA attack: We then perform TGDA attack using the updates in Section 2. We use total gradient ascent to update attacker parameter $\theta$ of L, and gradient descent to update defender parameter $\mathbf{w}$ of F.

(c) Label Information: We specify that $\mathcal{D}_p^0 = \{x_i, y_i\}_{i=1}^{\epsilon N}$. Prior works (e.g., [19, 23]) optimize $x$ to produce $x_p$, and perform a label flip on $y$ to produce $y_p$. This approach neglects label information during optimization. In contrast, we **fix** $y_p = y$, and concatenate $x$ and $y$ to $\mathcal{D}_p^0 = \{x_i; y_i\}_{i=1}^{\epsilon N}$ as input to L. Thus we generate poisoned points by considering the label information. We emphasize that we do not optimize or change the label during the attack, but merely use it to aid the construction of the poisoned $x_p$. Thus, our attack belongs to the clean label category.

*Testing*: Finally, we discuss how we measure the effectiveness of an attack. In a realistic setting, the testing procedure should be identical to the pretrain procedure, such that we can measure the effectiveness of $\mathcal{D}_p$ fairly. The consistency between pretrain and testing is crucial as the model F is likely to underfit with fewer training steps.

# 4 Experiments

We evaluate our TGDA attack on various models for image classification tasks and show the efficacy of our method for poisoning neural networks. Specifically, our results confirm that: (1) By applying the Stackelberg game formulation and incorporating second-order information, we can attack neural networks with improved efficiency and efficacy using the TGDA attack; (2) The efficient attack architecture further enables the TGDA attack to generate $\mathcal{D}_p$ in batches; (3) The poisoned points are visually similar to clean data, making the attack intuitively resistant to defenses.

## 4.1 Experimental Settings

**Dataset:** We consider image classification on the MNIST [5] (60,000 training and 10,000 test images), and CIFAR-10 [20] (50,000 training and 10,000 test images) datasets. We are not aware of prior work

Table 1: The attack accuracy/accuracy drop (%) and attack running time (hours) on the MNIST dataset. We only record the attack running time since pretrain and testing time are fixed across different methods. As the label flip attack does not involve optimization, its running time is always 0. Our attack outperforms the Back-gradient attack in terms of both effectiveness and efficiency across three different models.

| Target Model | Clean | Label Flip | | Back-gradient | | TGDA(ours) | |
|---|---|---|---|---|---|---|---|
| | Accuracy | Accuracy/Drop | Running time | Accuracy/Drop | Running time | Accuracy/Drop | Running time |
| LR | 92.35 | 90.83 / 1.52 | 0 hrs | 89.82 / 2.53 | 27 hrs | 89.56 / **2.79** | 1.1 hrs |
| NN | 98.04 | 97.99 / 0.05 | 0 hrs | 97.67 / 0.37 | 239 hrs | 96.54 / **1.50** | 15 hrs |
| CNN | 99.13 | 99.12 / 0.01 | 0 hrs | 99.02 / 0.09 | 2153 hrs | 98.02 / **1.11** | 75 hrs |

Table 2: The attack accuracy/accuracy drop (%) and attack running time (hours) on CIFAR-10.

| | Clean | Label Flip | MetaPoison | TGDA(ours) |
|---|---|---|---|---|
| Accuracy/Drop | 69.44 | 68.99 / 0.45 | 68.14/1.13 | 65.15 / **4.29** |
| Running time | 0 hrs | 0 hrs | 75hrs | 346 hrs |

that performs indiscriminate data poisoning on a dataset more complex than MNIST or CIFAR-10, and, as we will see, even these settings give rise to significant challenges in designing efficient and effective attacks. Indeed, some prior works consider only a simplified subset of MNIST (e.g., binary classification on 1's and 7's, or subsampling the training set to 1,000 points) or CIFAR-10 (e.g., binary classification on dogs and fish). In contrast, we set a benchmark by using the full datasets for multiclass classification.

**Baselines:** Among existing attacks, only a few can be directly compared with our method due to their attack formulations. For instance, the Poison SVM [2] and KKT [19] attacks can only be applied to convex models for binary classification; the Min-max [19] and the Model targeted [33] attacks can be only applied to convex models. Thus we compare with two baseline methods that can attack neural networks: the Back-gradient attack [23] and the Label flip attack [2]. Moreover, it is also possible to apply certain targeted attack method (e.g., MetaPoison [17]) in the context of the indiscriminate attack. Thus we compare with MetaPoison on CIFAR-10 under our unified architecture.

See Appendix C for a detailed list of our experimental settings (hardware and package, model details, dataset split, pretrain hyperparameters and complete baseline methods).

## 4.2 Comparison with Benchmarks

**MNIST.** We compare our attack with the Back-gradient and the Label flip attacks with $\epsilon = 3\%$ on MNIST in Table 1. Since the Back-gradient attack relies on generating poisoned points sequentially, we cannot adapt it into our unified architecture and run their code directly for comparison. For the label flip attack, we flip the label according to the rule that $y \leftarrow 10 - y$ as there are 10 classes.

We observe that label flip attack, though very efficient, is not effective against neural networks. Although [23] shows empirically that the Back-gradient attack is effective when attacking subsets of MNIST (1,000 training samples, 5,000 testing samples), we show that the attack is much less effective on the full dataset. We also observe that the complexity of the target model affects the attack effectiveness significantly. Specifically, we find that neural networks are generally more robust against indiscriminate data poisoning attacks, among which, the CNN architecture is even more robust. Overall, our method outperforms the baseline methods across the three target models. Moreover, with our unified architecture, we significantly reduce the running time of poisoning attacks by more than an order of magnitude.

**CIFAR-10.** We compare our attack with the Label flip attack and the MetaPoison attack with $\epsilon = 3\%$ on CIFAR-10 in Table 2. We omit comparison with the Back-gradient attack as it is too computationally expensive to run on CIFAR-10. We observe that the TGDA attack is very effective at poisoning the CNN architecture, but the running time becomes infeasible on larger models (e.g., ResNet). Also, MetaPoison is a more efficient attack (meta-learning with 2 unrolled steps are much quick than calculating total gradient), but since its original objective is to perform targeted attack, its application on indiscriminate attack is not effective. Moreover, the difference between the efficacy of the TGDA attack on MNIST and CIFAR-10 suggests that indiscriminate attacks may be dataset dependent, with MNIST being harder to poison than CIFAR-10.

Table 3: Comparing the TGDA attack with different orders: attacker as the leader and defender as the leader in terms of test accuracy/accuracy drop(%). Attacks are more effective when the attacker is the leader.

| Target Model | Clean | Attacker as leader | Defender as leader |
|---|---|---|---|
| LR | 92.35 | 89.56 / **2.79** | 89.79 / 2.56 |
| NN | 98.04 | 96.54 / **1.50** | 96.98 / 1.06 |
| CNN | 99.13 | 98.02 / **1.11** | 98.66 / 0.47 |

Table 4: Comparing the TGDA attack with different formulations: non-zero-sum and zero-sum in terms of test accuracy/accuracy drop (%). The non-zero-sum formulation is more effective at generating poisoning attacks.

| Target Model | Clean | Non Zero-sum | Zero-sum |
|---|---|---|---|
| LR | 92.35 | 89.56 / **2.79** | 92.33 / 0.02 |
| NN | 98.04 | 96.54 / **1.50** | 98.07 / -0.03 |
| CNN | 99.13 | 98.02 / **1.11** | 99.55 / -0.42 |

### 4.3 Ablation Studies:

**Who acts first.** In Section 2, we assume that the attacker is the leader and the defender is the follower, i.e., that the attacker acts first. Here, we examine the outcome of reversing the order, where the defender acts first. Table 3 shows the comparison. We observe that across all models, reversing the order would cause a less effective attack. This result shows that even without any defense strategy, the target model would be more robust if the defender acts one step ahead of the attacker.

**Attack formulation.** In Section 2, we discuss a relaxed attack formulation, where $\ell = f$ and the game is zero-sum. We perform experiments on this setting and show results in Table 4. We observe that the non-zero-sum formulation is significantly more effective, and in some cases, the zero-sum setting actually *increases* the accuracy after poisoning. We also find that using target parameters would not work for neural networks as they are robust to label flip attacks even when $\epsilon$ is large. We ran a label flip attack with $\epsilon = 100\%$ and observed only 0.1% and 0.07% accuracy drop on NN and CNN architectures, respectively. This provides further evidence that neural networks are robust to massive label noise, as observed by [26].

### 4.4 Visualization of attacks

We visualize some poisoned points $\mathcal{D}_p$ generated by the TGDA attack in Figure 2. The poisoned samples against NN and CNN are visually very similar with $\mathcal{D}_{tr}$, which provides heuristic evidence that the TGDA attack may be robust against data sanitization algorithms. We further quantitatively evaluate the robustness of TGDA attack agaisnt data sanitization algorithms in Appendix C.

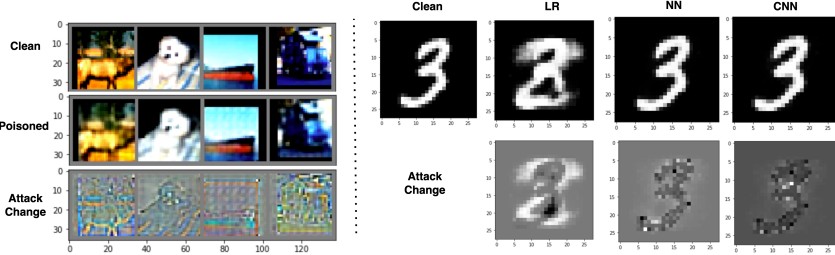

Figure 2: We visualize the poisoned data generated by the TGDA attack (left: CIFAR-10; right: MNIST).

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

| | Targeted Attacks | Backdoor Attacks | Indiscriminate Attacks | |
|---|---|---|---|---|
| **Objective** | Misclassifying target set of images (e.g. misclassifying car as cat(base class)) | Misclassifying images with the trigger pattern (e.g. misclassifying images with the white dot as cat) | Misclassifying every image (i.e. reducing the overall test accuracy) | **Perfect knowledge attack:** access to the training data, testing data and the training procedure (e.g. model target attack or any attack base on target parameters). |
| **Misclassify** | cat   cat   cat

cat   cat   cat | cat   cat   cat

cat   cat   cat | dog   cat  horse

truck  dog  ship | **Training only attack:** access to the training data and the training procedure (e.g. bi-level optimization methods). |
| **Correctly Classify** | All the other "untargeted" images | Images without a trigger | Images the attacker fail to misclassify | **Training data only attack:** access to the training data (e.g. label flip attacks) |

Figure 3: A taxonomy of data poisoning attacks for image classification. According to different objectives, *Target attacks* aim to misclassify a specific target set of images, *backdoor attacks* aim to misclassify any image with a specific trigger, and indiscriminate attacks aim at reducing the test accuracy in general. Among indiscriminate attacks, we can also identify three types of attacks according to the adversary's capability.

## A  Background on Data Poisoning Attacks

In this appendix, we categorize existing data poisoning attacks according to the attacker's *power* and *objectives*, and specify the type of attack we study in this paper.

### A.1  Power of an attacker

**Injecting poisoned samples.** Normally, without breaching one's database (i.e., changing the existing training data $\mathcal{D}_{tr}$), an attacker can only *inject* poisoned data, actively or passively to the defender's database, such that its objective can be achieved when the model is retrained after collection. Where the goal of the attacker can be presented as:

$$\mathbf{w}_* = \mathbf{w}_*(\mathcal{D}_p) \in \arg\min_{\mathbf{w}} \ \mathcal{L}(\mathcal{D}_{tr} \cup \mathcal{D}_p, \mathbf{w}), \tag{9}$$

where $\mathbf{w}_*$ is the desired model parameter, which realizes its respective objectives. We focus on such attacks and further categorize such objectives in the next subsection.

**Perturbing training data $\mathcal{D}_{tr}$.** An attacker may change $\mathcal{D}_{tr}$ directly on one occasion, where the attacker is also the "defender". Such an attacker can be the dataset owner, who intends to preserve a proprietary dataset to release but prevents others from benefiting by training it [16, 38, 10, 9], thus allows modification up to the entire $\mathcal{D}_{tr}$ to make it unlearnable.

For simplicity, we refer to injecting poisoned samples as *data poisoning attacks*, and perturbing training data as *dataset protection* in this paper.

### A.2  Objective of an attacker

Data poisoning attacks can be further classified into three categories according to the adversary's objective [12]. See Figure 3 for a taxonomic illustration.

**Targeted attack.** The attacker adds poisoned data $\mathcal{D}_p$ and acquire $\mathbf{w}^*$ such that a particular target example from the test set is misclassified as the *base* class [29, 1, 15, 40].This topic is well studied in the literature, and we refer the reader to [28] for an excellent summary of existing methods.

**Backdoor attack.** This attack aims at misclassifying any test input with a particular trigger pattern [14, 35, 4, 27]. Note that backdoor attacks require access to the input during inference time to plant the trigger.

**Indiscriminate attack.** This attack aims to acquire $\mathbf{w}^*$ by injecting poisoned data which decreases the model accuracy overall (or deny of service). We consider image classification tasks where

Table 5: Summary of existing poisoning attack algorithms. While some papers may include experiments on other datasets, we only cover vision datasets as our main focus is image classification. The attacks: Random label flip and Adversarial label flip attacks [2], P-SVM: PoisonSVM attack [2], Min-max attack [32], KKT attack [19], i-Min-max: improved Min-max attack [19], MT: Model Targeted attack [33], BG: Back-gradient attack [23].

| Attack | Dataset | Model | $|\mathcal{D}_{tr}|$ | $|\mathcal{D}_{test}|$ | $\epsilon$ | Code | Multiclass | Batch |
|---|---|---|---|---|---|---|---|---|
| Random label flip | toy | SVM | / | / | 0-40% | git | ✓ | $\epsilon|\mathcal{D}_{tr}|$ |
| Adversarial label flip | toy | SVM | / | / | 0-40% | git | ✗ | $\epsilon|\mathcal{D}_{tr}|$ |
| P-SVM | MNIST-17 | SVM | 100 | 500 | 0-9% | git | ✗ | 1 |
| Min-max | MNIST-17/Dogfish | SVM | 60000 | 10000 | 0-30% | git | ✓ | 1 |
| KKT | MNIST-17/Dogfish | SVM/LR | 13007/1800 | 2163/600 | 3% | git | ✗ | 1 |
| i-Min-max | MNIST | SVM | 60000 | 10000 | 3% | git | ✓ | 1 |
| MT | MNIST-17/Dogfish | SVM/LR | 13007/1800 | 2163/600 | / | git | ✓ | 1 |
| BG | MNIST | SVM, NN | 1000 | 8000 | 0-6% | git | ✓ | 1 |

the attacker aims to reduce the overall classification accuracy. Existing methods make different assumptions on the attacker's knowledge:

- Perfect knowledge attack: the attacker has access to both training and test data ($\mathcal{D}_{tr}$ and $\mathcal{D}_{test}$), the target model, and the training procedure (e.g., the min-max attack of [19]).
- Training-only attack: the attacker has access to training data $\mathcal{D}_{tr}$, the target model, and the training procedure (e.g., [23, 3]).
- Training-data-only attack: the attacker only has access to the training data $\mathcal{D}_{tr}$ (e.g., the label flip attack of [2]).

In this work, we focus on training-only attacks because perfect knowledge attacks are not always feasible due to the proprietary nature of the test data, while existing training-data-only attacks are weak and often fail for deep neural networks, as we show in Section 4.

# B    Experimental Protocol

We first summarize existing indiscriminate data poisoning attacks in Table 5, where we identify that such attacks work under subtly different assumptions, on, for example, the attacker's knowledge and the training set size. These inconsistencies result in unfair comparisons between methods .

# C    Additional Experiments

## C.1    Experimental Settings

**Hardware and package:** Experiments were run on a cluster with `T4` and `P100` GPUs. The platform we use is PyTorch. Specifically, autodiff can be easily implemented using `torch.autograd`. As for the total gradient calculation, we follow [39] and apply conjugate gradient for calculating Hessian-vector products.

**Dataset:** We consider image classification on the MNIST [5] (60,000 training and 10,000 test images), and CIFAR-10 [20] (50,000 training and 10,000 test images) datasets. We are not aware of prior work that performs indiscriminate data poisoning on a dataset more complex than MNIST or CIFAR-10, and, as we will see, even these settings give rise to significant challenges in designing efficient and effective attacks. Indeed, some prior works consider only a simplified subset of MNIST (e.g., binary classification on 1's and 7's, or subsampling the training set to 1,000 points) or CIFAR-10 (e.g., binary classification on dogs and fish). In contrast, we set a benchmark by using the full datasets for multiclass classification.

**Training and validation set:** During the attack, we need to split the clean training data to the training set $\mathcal{D}_{tr}$ and validation set $\mathcal{D}_v$. Here we split the data to 70% training and 30% validation, respectively.

Table 6: Comparison with pGAN on MNIST with loss defense.

| Method | TGDA (w/wo defense) | | | pGAN(w/wo defense) | | |
|---|---|---|---|---|---|---|
| Target Model | LR | NN | CNN | LR | NN | CNN |
| Accuracy Drop (%) | 2.79/2.56 | 1.50/1.49 | 1.11/1.104 | 2.52/2.49 | 1.09/1.07 | 0.74/0.73 |

Table 7: TGDA attack on MNIST with MaxUp defense.

| Method | TGDA (w/wo defense) | | |
|---|---|---|---|
| Target Model | LR | NN | CNN |
| Accuracy Drop (%) | 2.79/2.77 | 1.50/1.50 | 1.11/1.11 |

Thus, for the MNIST dataset, we have $|\mathcal{D}_{tr}| = 42000$ and $|\mathcal{D}_v| = 18000$. For the CIFAR-10 dataset, we have $|\mathcal{D}_{tr}| = 35000$ and $|\mathcal{D}_v| = 15000$.

**Attacker models and Defender models:** (1) For the attacker model, for MNIST dataset: we use a three-layer neural network, with three fully connected layers and leaky ReLU activations; for CIFAR-10 dataset, we use an autoencoder with three convoluational layers and three conv transpose layers. The attacker takes the concatenation of the image and the label as the input, and generates the poisoned points. (2) For the defender, we examine three target models for MNIST: Logistic Regression, a neural network (NN) with three layers and a convolutional neural network (CNN) with two convolutional layers, maxpooling and one fully connected layer; and only the CNN model for CIFAR-10 (as CIFAR-10 contains RBG images).

**Hyperparameters:** (1) Pretrain: we use a batch size of 1,000 for MNIST and 256 for CIFAR-10, and optimize the network using our own implementation of gradient descent with `torch.autograd`. We choose the learning rate as 0.1 and train for 100 epochs. (2) Attack: for the attacker, we choose $\alpha = 0.01$, $m = 1$ by default; for the defender, we choose $\beta = 0.1$, $n = 20$ by default. We set the batch size to be 1,000 for MNIST; 256 for CIFAR10 and train for 200 epochs, where the attacker is updated using total gradient ascent and the defender is updated using gradient descent. We follow [39] and implement TGA using conjugate gradient. We choose the poisoning fraction $\epsilon = 3\%$ by default. Note that choosing a bigger $\epsilon$ will not increase our running time, but we choose a small $\epsilon$ to resemble the realistic setting in which the attacker is limited in their access to the training data. (3) Testing: we choose the exact same setting as pretrain to keep the defender's training scheme consistent.

**Baselines:** There is a spectrum of data poisoning attacks in the literature. However, due to their attack formulations, only a few attacks can be directly compared with our method. For instance, the Poison SVM [2] and KKT [19] attacks can only be applied to convex models for binary classification; the Min-max [19] and the Model targeted [33] attacks can be only applied to convex models. Thus we compare with two baseline methods that can attack neural networks: the Back-gradient attack [23] and the Label flip attack [2]. Moreover, it is also possible to apply certain targeted attack method (e.g., MetaPoison [17]) in the context of the indiscriminate attack. Thus we compare with MetaPoison on CIFAR-10 under our unified architecture. We follow [17] and choose $K = 2$ unrolled inner steps, 60 outer steps and an ensemble of 24 inner models.

## C.2 Against Defenses:

To further evaluate the robustness of TGDA attack against data sanitization algorithms:

(1) We perform the loss defense [19] by removing 3% of training points with the largest loss. We compare with pGAN [24], which include constraint on the similarity between clean and poisoned sample thus inherently robust against defenses. In Table 6, we observe that although we do not add explicit constraint on detectability in our loss function, our method still reaches comparable robustness against such defenses with pGAN.

(2) We examine the robustness of our TGDA attack against strong data augmentations, e.g., the MaxUp defense[3] [13]. In a nutshell, MaxUp generates a set of augmented data with random

---

[3]We follow implementation in `https://github.com/Yunodo/maxup`

perturbations and then aims at minimizing the worst case loss over the augmented data. Such training technique addresses overfitting and serves as a valid defense against adversarial examples. However, it is not clear if MaxUp is a good defense against indiscriminate data poisoning attacks. Thus, we implement MaxUp under our testing protocol, where we add random perturbations to the training and the poisoned data, i.e., $\{\mathcal{D}_{tr} \cup \mathcal{D}_p\}$, and then minimize the worst case loss over the augmented set. We report the results in Table 7, where we observe that even though MaxUp is a good defense against adversarial examples, it is not readily an effective defense against indiscriminate data poisoning attacks. Part of the reason we believe is that in our formulation the attacker anticipates the retraining done by the defender, in contrast to the adversarial example setting.

