# OpenReview forum: "Indiscriminate Data Poisoning Attacks on Neural Networks"
_NeurIPS.cc/2022/Workshop/TSRML — TSRML2022_

### Official Review · Reviewer_ujvr · 2022-10-20

**Overall Recommendation:** See above.
**Overall Rating:** 7

**Summary:**

This paper studies indiscriminate data poisoning attacks on neural networks by formulating them as a Stackelberg game.

**Strengths:**

+ A new method to solve the bi-level optimization problem is designed.
+ The proposed method is compared with existing studies, including both targeted and untargeted attacks.

**Weaknesses:**

- Even though the proposed attack is better than existing methods, the attack is still not very effective.
- Impact of $\epsilon$ is not studied.
-The computation cost is large.

**Review Confidence:**

3: The reviewer is fairly confident that the evaluation is correct

---

### Official Review · Reviewer_G3hX · 2022-10-20
**Limited contribution&evaluation**

**Overall Rating:** 6

**Summary:**

This paper proposes
1. using 'total gradient descent ascent' to optimize poisoned samples
2. using an autoencoder to model the generation of poisoned samples

as ways to improve effectiveness and efficiency of indiscriminate data poisoning attacks (i.e. where the adversary inject poisoned samples to reduce overall model accuracy).


**Strengths:**

1. The authors explain in details the motivations behind their evaluation protocols, which should be appreciated.
2. The paper is overall clear.
3. The proposed method of fine-tuning a autoencoder to generate poison samples is reasonable.

**Weaknesses:**

1. The novelty is limited.
The non-zero sum formulation (i.e. the equation 5) is not new, as [23,17,19] cited in the paper;
The 'total gradient descent ascent' is adapted from [7] & [8].

2. Only a few baselines are evaluated. I understand that the scope is restricted to indiscriminate data poisoning attacks only rather than targeted/untargeted/backdoor attacks, but many poisoning attacks can trivially applied to indiscriminate data poisoning as well. Many can be found in the survey[12] that is already cited in the paper.

3. The claim that the proposed scheme is robust against data sanitization algorithms is not sufficiently supported by the evidence in Appendix C.2. Only two simple defenses are evaluated: One of them (MaxUp) are not even proposed for defending poisoning attacks; The other one (loss defense) is a simple baseline defense that filtering out samples with the largest loss values, and only a simple setting (removing 3%, exactly the same as the attack budget) is reported.

**Overall Recommendation:**

I see no obvious factual error but the evaluation seems inadequate to support their claim regarding the effectiveness of their attack .

**Review Confidence:**

4: The reviewer is confident but not absolutely certain that the evaluation is correct

---

### Decision · Program_Chairs · 2022-10-23

Accept